

# Genome-wide identification of the trehalose-6-phosphate synthase gene family in sweet orange (*Citrus sinensis*) and expression analysis in response to phytohormones and abiotic stresses

Kehong Liu and Yan Zhou

National Citrus Engineering Research Center, Citrus Research Institute, Southwest University, Chongqing, China

## ABSTRACT

**Background**. Trehalose-6-phosphate synthase (TPS) is an essential enzyme for synthesizing trehalose and is a significant regulator of plant development and stress response. Sweet orange (*Citrus sinensis*) is an economically important fruit tree crop and a common transgenic material. At present, little information is available about the *TPS* gene family in sweet orange.

**Methods**. The *TPS* gene family were identified from sweet orange genome by bioinformatics analysis. Additionally, the expression of *CisTPS* genes was analyzed under phytohormones and abiotic stresses by quantitative reverse-transcriptase polymerase chain reaction (qRT-PCR).

**Results**. Here, eight *TPS* genes were identified and were found to be randomly distributed in five sweet orange chromosomes. TPS and trehalose-6-phosphate phosphatase (TPP) domains were observed in all CisTPS proteins. The phylogenetic tree showed that *CisTPS* genes were divided into two subfamilies, and genes in each subfamily had conserved intron structures and motif compositions. The *cis*-acting elements of *CisTPS* genes suggested their roles in phytohormone and stress responses. All *CisTPS* genes were ubiquitously expressed in roots, leaves, and stems, and six members were highly expressed in roots. Expression profiles showed that *CisTPS* genes exhibited tissue specificity and were differentially expressed in response to phytohormones and abiotic stresses. This study lays a foundation for revealing the functions of the *TPS* gene family in trehalose regulation in sweet orange, and provides a valuable reference for this gene family in other plants.

## INTRODUCTION

Trehalose is a non-reducing disaccharide composed of two glucose units connected by an alpha,alpha-1,1-glycosidic linkage (*Elbein et al., 2003*; *Bansal et al., 2013*), and widely found in bacteria, fungi, slime molds, protozoa, invertebrates, and higher plants (*Becker et al., 1996*; *Bansal et al., 2013*; *Lunn et al., 2014*; *Tang et al., 2018*). Trehalose metabolism is

Corresponding author
Kehong Liu, liukehong@cric.cn

involved in growth, development, and abiotic stress response in higher plants (*Kosar et al., 2019*; *Paul, Watson & Griffiths, 2020*).

Thus far, five trehalose biosynthetic pathways have been identified, including trehalose-6-phosphate synthase (TPS)/trehalose-6-phosphate phosphatase (TPP), TreY/TreZ, TreS, TreP, and TreT pathways; however, only the TPS/TPP pathway is found in higher plants (*Avonce et al., 2006*; *Paul et al., 2008*; *Lunn et al., 2014*). The TPS/TPP pathway involves a two-step reaction. First, catalyzed by TPS, trehalose-6-phosphate (T6P) is produced from UDP glucose and glucose-6-phosphate. Second, catalyzed by TPP, T6P is converted to trehalose (*Cabib & Leloir, 1958*; *Goddijn & Van Dun, 1999*). Thus, TPS is an essential enzyme for trehalose synthesis in the TPS/TPP pathway.

In higher plants, the *TPS* gene family is divided into two distinct classes—Class I and II (*Lunn, 2007*), which differ in gene expression pattern, enzyme activity, and physiological function (*Ping et al., 2019*). Only Class I members encoding catalytically active enzymes have TPS activity (*Blázquez et al., 1998*; *Vandesteene et al., 2010*), whereas Class II members lack TPS and TPP activity and their functions remain unclear (*Ramon et al., 2009*; *Lunn et al., 2014*). The *TPS* gene family is a small gene family, where the number of members varies among species (*Wei et al., 2016*). For example, there are 11 members in *Arabidopsis thaliana*, rice, and pepper (*Leyman, Dijck & Thevelein, 2001*; *Zang et al., 2011*; *Wei et al., 2016*), 12 in winter wheat and poplar (*Yang et al., 2012*; *Xie et al., 2015a*), and seven in grapevine and cucumber (*Dan et al., 2021*; *Morabito, Secchi & Schubert, 2021*). Most TPS proteins contain both conserved TPS and TPP domains, and a few TPS proteins only contain the TPS domain (*Yang et al., 2012*; *Lin et al., 2018*; *Sun, Chen & Tao, 2021*).

The *TPS* gene family also plays a vital role in plant embryo development, flower induction, senescence regulation, seed filling, and biotic and abiotic stress tolerance in plants (*Gómez et al., 2010*; *Wingler et al., 2012*; *Wahl et al., 2013*; *Kumar et al., 2019*; *Zhao et al., 2019*). For instance, the *AtTPS1* gene is a regulator of glucose, abscisic acid (ABA), and stress signaling (*Avonce et al., 2004*). The *AtTPS1* null mutant showed arrested embryo development, hindered vegetative growth, and delayed flowering (*Eastmond et al., 2002*; *Gómez, Baud & Graham, 2005*; *Gómez et al., 2010*). *AtTPS1* overexpression can enhance drought resistance in *A. thaliana* (*Avonce et al., 2004*). Overexpressing the gene encoding the bifunctional fusion of *TPS* and *TPP* genes from *Escherichia coli* in transgenic tomato plants improved drought and salt resistance and photosynthetic rates (*Lyu et al., 2013*).

*OsTPS1* overexpression enhanced tolerance to stresses such as salt, drought, and low temperature in transgenic rice by increasing the trehalose and proline content and regulating the expression of stress-related genes. Furthermore, *OsTPS1*-overexpressed transgenic rice did not cause any clear phenotypic changes (*Li et al., 2011*). *SlTPS1* of *Selaginella lepidophylla* is involved in the response to heat and salinity by enhancing T6P biosynthesis (*Zentella et al., 1999*). *AtTPS5* participates in the regulation of heat shock response by interacting with MBF1c and is a negative regulator in ABA signal transduction (*Suzuki et al., 2008*; *Tian et al., 2019*). *AtTPS6* can control plant architecture, epidermal pavement cell shape, and trichome branching (*Chary et al., 2008*).

Sweet orange (*Citrus sinensis*) is an economically important fruit tree crop and a common transgenic material. The *TPS* gene family has been functionally and phylogenetically

characterized in the model plant *A. thaliana* (*Vandesteene et al., 2010*), important cash crops (rice, cotton, potato, and soybean) (*Zang et al., 2011*; *Xie, Wang & Huang, 2014*; *Mu et al., 2016*; *Xu et al., 2017*), horticultural plants (tree peony and petunia) (*Dong et al., 2019*; *Sun, Chen & Tao, 2021*), and woody plants (poplar and apple) (*Yang et al., 2012*; *Du et al., 2017*). However, information about the *TPS* gene family in sweet orange is scarce. In this study, we predicted the *TPS* genes in sweet orange based on sweet orange genomic sequences, and analyzed the gene structure, chromosomal location, motif distribution, phylogenetic relationship, and expression patterns by bioinformatics methods. These findings lay a foundation for future research on the functions of *TPS* genes in sweet orange and will contribute to the genetic improvement of citrus.

## MATERIALS & METHODS

### Identification of *TPS* gene family in sweet orange

The candidate TPS protein sequences in sweet orange (*C. sinensis*) were downloaded from Phytozome v13 (https://phytozome-next.jgi.doe.gov) (*Goodstein et al., 2012*). Then, the TPS (Glyco-transf-20, PF00982) and TPP (Trehalose_PPase, PF02358) domains were predicted using the SMART website (http://smart.embl-heidelberg.de) and the National Center for Biotechnology Information Conserved Domain Database (NCBI-CDD; https://www.ncbi.nlm.nih.gov/cdd) (*Lu et al., 2020*; *Letunic, Khedkar & Bork, 2021*), and proteins lacking the TPS domain were removed.

The TPS cDNA sequences were used as queries to search the *C. sinensis* genome database at NCBI to confirm the chromosome localization of *TPS* genes. The *TPS* genes were named based on their location on *C. sinensis* chromosomes, and their physical locations were visualized using MG2C_v2.1 (http://mg2c.iask.in/mg2c_v2.1). The basic information on CisTPS proteins, including molecular weight (MW), isoelectric point (pI), grand average of hydropathicity (GRAVY), and subcellular locations were predicted using the Expasy (https://web.expasy.org/protparam/) and GenScript (https://www.genscript.com/wolf-psort.html) websites (*Duvaud et al., 2021*). The secondary structures of CisTPS proteins were predicted using the PRABI website (https://npsa-prabi.ibcp.fr/cgi-bin/npsa_automat.pl?page=npsa_sopma.html). The collinearity and selective evolutionary pressure of *TPS* genes were analyzed using the TBtools software (*Chen et al., 2020*).

### Phylogenetic analyses

Based on previous studies, 51 protein sequences were downloaded, including 10 PoTPS, 7 CsTPS, 12 PtTPS, 11 AtTPS and 11 OsTPS protein sequences (*Blázquez et al., 1998*; *Dan et al., 2021*; *Sun, Chen & Tao, 2021*; *Vogel et al., 2001*; *Yang et al., 2012*; *Zang et al., 2011*). Multiple alignments of CisTPS, PoTPS, CsTPS, PtTPS, AtTPS, and OsTPS protein sequences were performed using ClustalW, and the Neighbor-Joining (NJ) phylogenetic tree was constructed using MEGA-X with a 1000 bootstrap test.

### Gene structure and motif analyses

The gene structure of *CisTPS* genes was analyzed and visualized using GSDS v2.0 (http://gsds.gao-lab.org/) (*Hu et al., 2015*). The conserved motifs in the CisTPS proteins

were identified using MEME Suite v5.4.1 (https://meme-suite.org/meme/tools/meme) with the parameter settings: number of repetitions = any and maximum number of motifs = 20 (*Timothy et al., 2009*).

## Prediction of *cis*-acting elements

Using Phytozome v13 (https://phytozome-next.jgi.doe.gov), upstream sequences (2000 bp) of *CisTPS* genes were extracted from the sweet orange genome as promoter sequences. PlantCARE (http://bioinformatics.psb.ugent.be/webtools/plantcare/html/) was used to predict the *cis*-acting elements in the promoter sequences, and the results were illustrated with the TBtools software (*Lescot et al., 2002*; *Chen et al., 2020*).

## Plant materials and treatment conditions

The outer and inner seed coats of the Ridge pineapple sweet orange seeds were removed, and sterilized seeds were cultured in Murashige and Skoog (MS) solid medium in a light incubator (27 °C, 16 h light/8 h dark) for 30 d. The culture seedlings were used as test materials. Roots, leaves, and stems of seedlings were collected and stored at −80 °C to calculate the *CisTPS* gene expression in different tissues.

Seedlings were transferred to an MS liquid medium and placed at 27 °C as control. For temperature stress treatments, the seedlings in the MS liquid medium were placed at a high temperature (40 °C) or a low temperature (4 °C) (*Xie et al., 2018*). For phytohormone and abiotic stress treatments, seedlings were transferred to an MS liquid medium containing 100 μM ABA, 50 μM indole-3-acetic acid (IAA), 10% (w/v) polyethylene glycol (PEG-6000), and 150 mM NaCl, and placed at 27 °C (*Liu et al., 2020*; *Xie et al., 2015b*; *Xu et al., 2017*). Leaves were immediately frozen in liquid nitrogen and stored at −80 °C after each treatment at 0, 6, 12, and 24 h. Three independent biological replicates were performed, and the leaves of each sample were collected from a single seedling.

## Expression profile analysis

The specific primers for detecting *CisTPS* genes were designed by Primer Premier v6.0, and *FBOX* was the housekeeping gene used as an internal reference (*Mafra et al., 2012*). The primer sequences are listed in Table S1.

The total RNA was extracted with TRIzol Reagent (Invitrogen, Waltham, MA, USA), and first-strand cDNA was synthesized with 1 μg of total RNA using All-In-One 5×RT MasterMix (Applied Biological Materials Inc., Richmond, BC, Canada). Total RNA extraction and cDNA synthesis were performed according to the manufacturers' instructions. The synthesized cDNA solution was diluted 10 times with distilled water, and the diluted cDNA was used as a template for quantitative polymerase chain reaction (qPCR). qPCR was performed with TB Green® Premix Ex Taq™ II kit (TaKaRa, Beijing, China). The qPCR reaction mixture consisted of 9 μL template cDNA, 0.5 μL each of 10 μM primers, and 10 μL SYBR Green Supermix. qPCR was performed for 3 min at 95 °C (one cycle), followed by 10 s at 95 °C, 60 s at 60 °C (40 cycles). Each reaction was performed in technical triplicates.

Relative gene expression was calculated by the $2^{-\Delta\Delta Ct}$ method (*Livak & Schmittgen, 2001*). Standard error bars represent standard error of the mean (SEM). The expression

**Table 1  Summary of *CisTPS* genes.**

| Gene | Gene ID | Chromosome location | TPS domain location | TPP domain location | ORF/aa | MW/KD | PI | GRAVY | Location |
|------|---------|--------------------|--------------------|--------------------|--------|-------|------|--------|----------|
| *CisTPS1* | XM_006467507.3 | Chr2: 2018989-2028927 | 105-572 | 616-850 | 942 | 106.78 | 6.29 | −0.333 | Chloroplast |
| *CisTPS2* | XM_006467546.3 | Chr2: 2235199-2239556 | 61-546 | 595-830 | 856 | 96.24 | 5.96 | −0.202 | Chloroplast |
| *CisTPS3* | XM_015527352.2 | Chr2: 14383332-14388004 | 60-528 | 577-812 | 831 | 94.24 | 5.59 | −0.186 | Cytoplasm |
| *CisTPS4* | XM_006471525.3 | Chr3: 13930419-13936240 | 59-547 | 596-831 | 863 | 96.62 | 5.60 | −0.132 | Cytoplasm |
| *CisTPS5* | XM_006474056.3 | Chr4: 1295233-1298512 | 51-540 | 589-824 | 854 | 96.54 | 5.73 | −0.238 | Cytoplasm |
| *CisTPS6* | XM_006476690.3 | Chr5: 3082065-3088262 | 59-546 | 583-818 | 832 | 94.55 | 6.11 | −0.199 | Cytoplasm |
| *CisTPS7* | XM_006477757.3 | Chr5: 10415267-10433927 | 94-561 | 605-838 | 937 | 105.65 | 6.38 | −0.392 | Cytoplasm |
| *CisTPS8* | XM_006483756.3 | Chr7: 5247964-5251328 | 59-546 | 595-830 | 861 | 96.81 | 6.01 | −0.208 | Cytoplasm |

of *CisTPS* genes in different tissues was normalized by that in roots (*Dan et al., 2021*). Statistical differences were analyzed with Student's *t*-test.

# RESULTS

## Genome-wide identification of *TPS* genes in sweet orange

Eight *TPS* genes were identified in the sweet orange genome by bioinformatics analysis. Based on the assessment of Pfam and CDD, these eight TPS proteins contained two conserved domains-an N-terminal TPS domain (Glyco_transf_20; Pfam: PF00982) and a C-terminal TPP domain (Trehalose_PPase; Pfam: PF02358) (Table 1). These results confirmed that the eight genes belonged to the *TPS* gene family. The *TPS* genes were named *CisTPS1-CisTPS8* according to chromosome position (Fig. 1). Furthermore, CisTPS2-CisTPS5 proteins contained an extra Hydrolase_3 domain (Pfam: PF08282).

CisTPS genes were distributed on five chromosomes, three on chromosome 2, two on chromosome 5, and one on chromosomes 3, 4, and 7 (Fig. 1). The genes were mostly located at the proximal ends of chromosomes. No obvious correlation was observed between chromosome length and number of *CisTPS* genes based on their distribution on chromosomes.

Physicochemical properties analysis revealed that the size of CisTPS proteins was highly variable from 831 (CisTPS3) to 942 amino acids (CisTPS1), and MW was between 94.24 KDa and 106.78 KDa. pI ranged from 5.59 (CisTPS3) to 6.38 (CisTPS7). GRAVY was predicted from −0.392 (CisTPS7) to −0.132 (CisTPS4). Subcellular prediction of these *CisTPS* genes indicated their localization in the chloroplast and cytoplasm (Table 1).

Analysis of the secondary structure content in CisTPS proteins showed that these proteins consisted of alpha helix, beta turn, random coil, and extended strand (Table S2).

To better understand the evolutionary mechanism of sweet orange *TPS* family, a collinear relationship diagram of sweet orange, *A. thaliana* and rice was constructed. Ten pairs of orthologous genes were found between sweet orange and *A. thaliana*, and six between sweet orange and rice (Fig. 2). Furthermore, the orthologous genes of *CisTPS2*, *CisTPS5* and *CisTPS6* were detected in both dicotyledon (*A. thaliana*) and monocotyledon (rice) (Fig. 2), indicating the three genes may be highly conserved. The Ka /Ks ratios of all orthologous
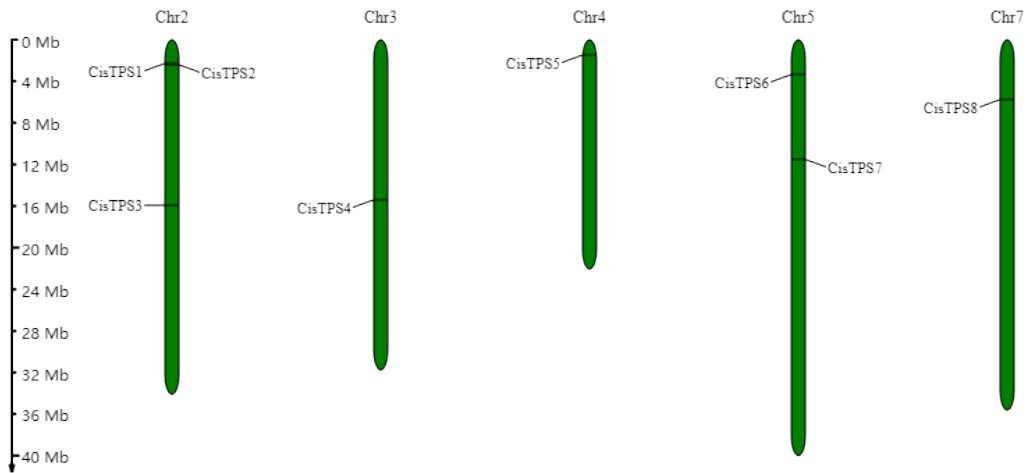

**Figure 1 Location of *CisTPS* genes in sweet orange genome.** The chromosome numbers are showed on the top of each bar and the left scale represents the megabases.

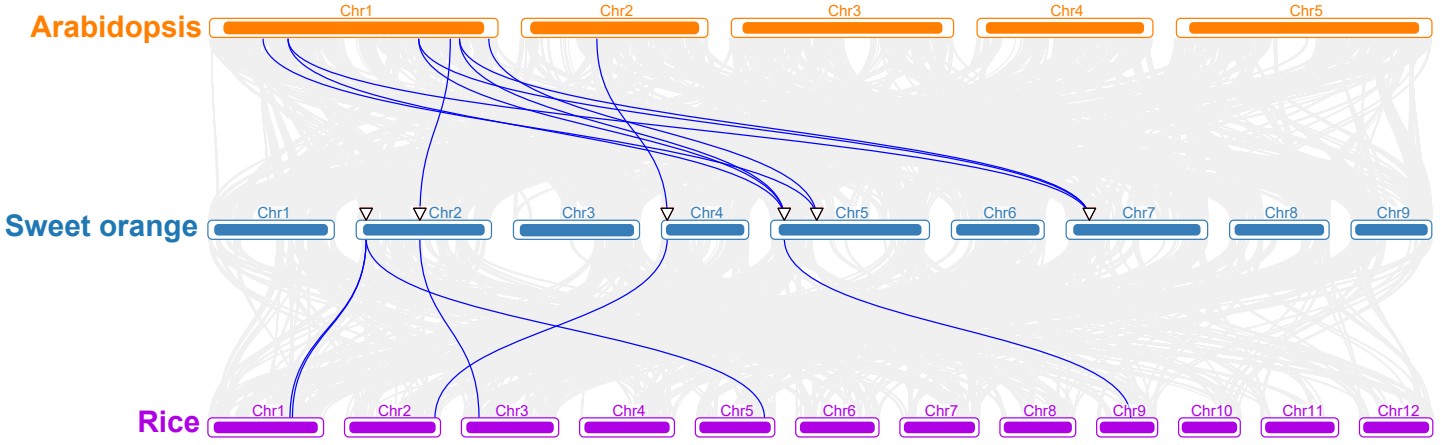

**Figure 2 Synteny analysis of *TPS* genes among *Arabidopsis*, sweet orang and rice.** Gray lines in the background indicate all the collinear blocks, and blue lines represent the syntenic *TPS* gene pairs.

genes between sweet orange and *A. thaliana* were less than 1(Table S3), suggesting purifying selection had acted upon these orthologous genes.

## Phylogenetic analysis of *CisTPS* gene family

To determine the evolutionary relationship of *TPS* genes among various species, a phylogenetic tree was constructed by the NJ method based on the TPS protein amino acid sequences from cucumber, tree peony, *Populus*, *A. thaliana*, rice, and sweet orange. *CisTPS* genes were classified into two subfamilies, Class I and Class II, as shown in the previous studies in cucumber, tree peony, *Populus*, *A. thaliana* and rice (Fig. 3) (*Blázquez et al., 1998*; *Dan et al., 2021*; *Sun, Chen & Tao, 2021*; *Yang et al., 2012*; *Vogel et al., 2001*; *Zang*

*et al., 2011*). *CisTPS1* and *CisTPS7* belonged to Class I, and the other genes belonged to Class II (Fig. 3).

Most *CisTPS* and *PtTPS* genes clustered together, suggesting that *CisTPS* genes were closely related to *PtTPS* genes. For example, *CisTPS1* clustered with *PtTPS2*, *CisTPS6* clustered with *PtTPS9* and *PtTPS10*, *CisTPS2* clustered with *PtTPS7* and *PtTPS8*, and *CisTPS3* clustered with *PtTPS4* and *PtTPS6*.

## Structure analysis of *CisTPS* genes

To better understand the molecular characteristics of *CisTPS* genes, the gene structures such as exons, introns, and conserved motifs were analyzed. In Class I, *CisTPS1* and *CisTPS7* contained 18 and 17 exons, and 17 and 16 introns, respectively. However, in Class II, *CisTPS3* and *CisTPS6* possessed four exons and three introns, whereas all other genes contained three exons and two introns (Fig. 4B). The results indicated that functional diversity of closely related *TPS* genes might be caused by the gain and loss of exons in the course of evolution of the *TPS* gene family. In addition, 20 distinct conserved motifs were searched using the MEME website. The lengths of these conserved motifs ranged from 15 to 50 amino acids (Table S4). Members (*CisTPS1* and *CisTPS7*) of Class I all harbored 15 motifs, which lacked motif 8, 10, 15, 19, and 20. Members of Class II contained all 20 motifs, except for *CisTPS5*, which contained 19 motifs but lacked motif 20 (Fig. 4C). The results of the structure analysis confirmed the reliability of the phylogenetic tree (Fig. 4A), suggesting functional differences between Class I and II.

## *Cis*-acting elements of *CisTPS* genes

By analyzing the 2,000-bp region upstream of the transcription start site, 70 types of *cis*-acting elements were discovered in the promoter regions of *CisTPS* genes (Table S5). Among these, seven types of *cis*-acting elements were related to abiotic stress and nine types of *cis*-acting elements were related to phytohormone responses. In abiotic stress-related elements, ARE (an anaerobic induction element) and MBS (a drought-inducible element) were mainly found in the promoter regions of *CisTPS* genes. Phytohormone-responsive elements, including ABRE (ABA-responsive element), CGTCA-motif (methyl jasmonic acid [MeJA]-responsive element), TGACG-motif (MeJA-responsive element), and TGA-element (auxin-responsive element), were observed in most *CisTPS* genes. A total of 126 *cis*-elements related to light were recognized on the promoters, and each promoter harbored at least nine light-responsive elements (Fig. 5). In addition, the CAAT-box and TATA-box, which are considered the core and common promoter elements, were found in the promoter regions of all *CisTPS* genes. *CisTPS* genes may play essential roles in response to abiotic stresses, phytohormones, and light.

## Expression analysis of *CisTPS* genes in different tissues

To determine the specificity of *TPS* gene expression in sweet orange, *CisTPS* gene expression in roots, stems, and leaves was quantified using qRT-PCR. All *CisTPS* genes were expressed in roots, stems, and leaves, and the expression levels of most *CisTPS* genes were lower in leaves than in other tissues. *CisTPS2*, *CisTPS3*, *CisTPS4*, *CisTPS6*, *CisTPS7* and *CisTPS8*

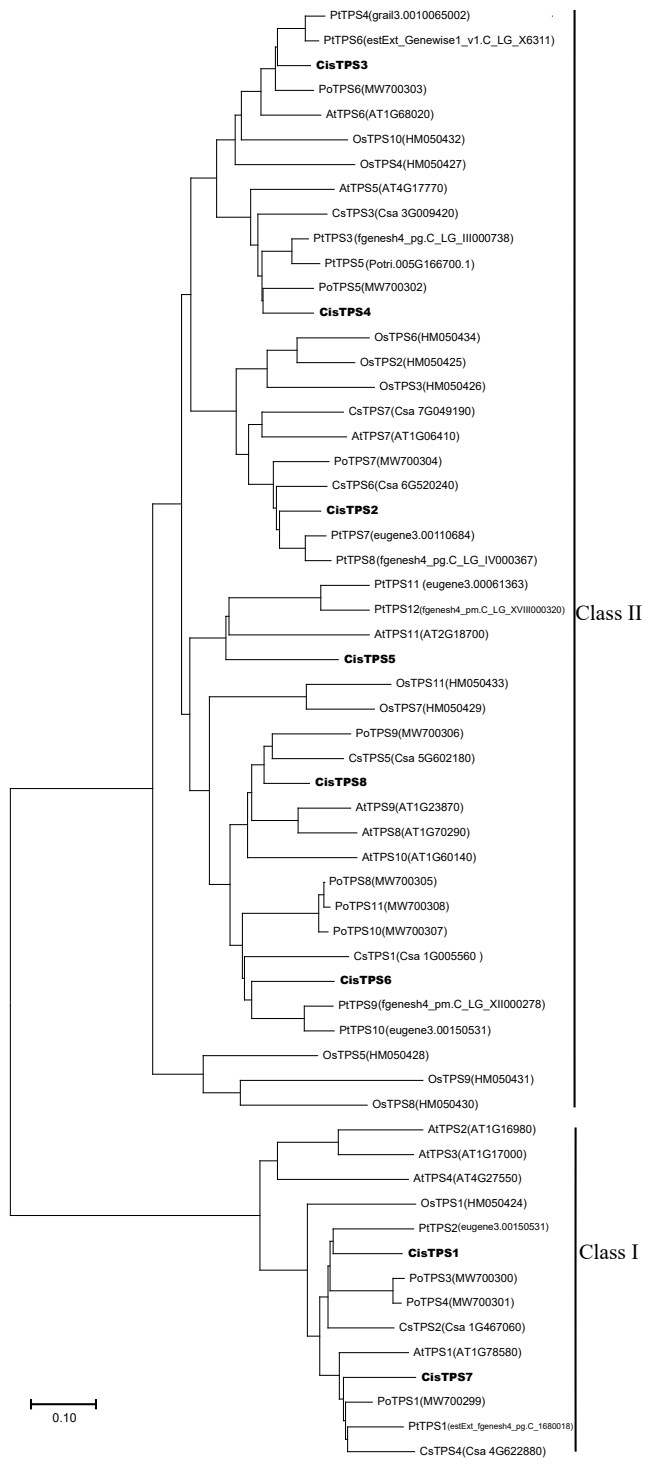

**Figure 3** **Phylogenetic relationship of TPS proteins among sweet orang,** *Arabidopsis,* **rice, cucumber,** *populus* **and tree peony.** The phylogenetic tree was constructed using the Neighbor-Joining method by MEGA-X with 1 000 bootstrap replicates.

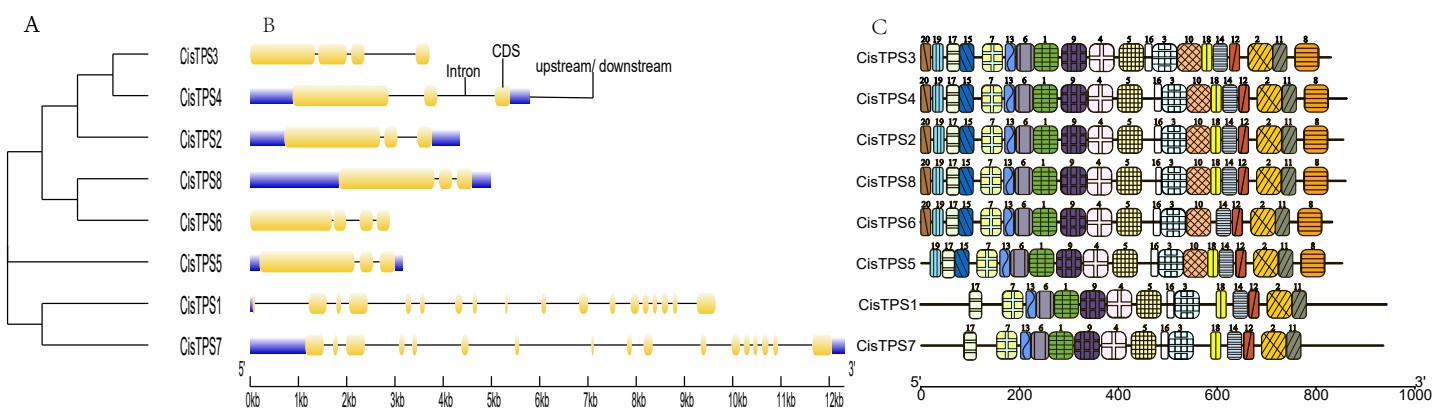

**Figure 4** **Phylogenetic relationship, gene structure and motif compositions of *CisTPS* genes.** (A) A phylogenetic tree of eight CisTPS proteins. (B) Exon-intron structure of *CisTPS* genes. (C) Conserved motifs of eight CisTPS proteins.

were highly expressed in roots, whereas *CisTPS1* and *CisTPS5* were highly expressed in stems (Fig. 6).

**Expression analysis of *CisTPS* genes under phytohormone treatment**
TPS proteins are involved in the differential regulation of gene expression. To understand the potential roles of *CisTPS* genes, we measured their expression characteristics in sweet orange seedlings after phytohormone treatment. Under ABA treatment, the expression of *CisTPS3* and *CisTPS7* was slightly upregulated at 24 and 6 h, respectively. *CisTPS4* was upregulated and peaked at 6 h, declined to its lowest expression level at 12 h, and recovered to an intermediate level at 24 h. *CisTPS5* and *CisTPS8* were slightly downregulated, with the lowest expression at 12 and 24 h, respectively. Under IAA treatment, *CisTPS2* and *CisTPS7* were upregulated at 6 and 12 h, respectively, *CisTPS3* at 12 h, and *CisTPS8* at 6 h, whereas *CisTPS4* was slightly inhibited at 24 h (Fig. 7).

**Expression analysis of *CisTPS* genes under abiotic treatment**
We also examined the expression of *CisTPS* genes in response to various abiotic stresses, and qRT-PCR was used to calculate the expression patterns under different treatment conditions. Under NaCl treatment, *CisTPS2* and *CisTPS3* were significantly induced at 12 h. However, *CisTPS7* was significantly upregulated at 12 h and slightly repressed at 24 h. Under PEG-6000, *CisTPS1* showed strong expression at 6 h, and returned to normal levels after 6 h. At 24 h, *CisTPS2* expression was slightly suppressed, whereas *CisTPS3* and *CisTPS7* expression was slightly increased. *CisTPS2*, *CisTPS3*, *CisTPS4*, and *CisTPS7* were upregulated at low temperature; *CisTPS2* and *CisTPS7* were strongly expressed at both 12 and 24 h. *CisTPS1* and *CisTPS4* were significantly upregulated at 12 and 6 h at 40 °C treatment conditions, respectively. *CisTPS7* was slightly induced by high temperature at 6 h (Fig. 8).

| CisTPS1 | CisTPS2 | CisTPS3 | CisTPS4 | CisTPS5 | CisTPS6 | CisTPS7 | CisTPS8 | Element | Category |
|---|---|---|---|---|---|---|---|---|---|
| 5 | 2 | 3 | 5 | 6 | 1 | 3 | 2 | Box 4 | Light response |
| 2 | 1 | 1 |  |  | 3 |  | 1 | GT1-motif | |
| 1 |  | 1 | 1 |  |  |  |  | MRE | |
| 1 |  |  |  |  |  |  |  | Sp1 | |
| 5 | 14 | 8 |  | 7 | 5 | 3 | 6 | G-Box | |
| 1 |  |  |  |  |  |  |  | Gap-box | |
| 1 | 1 | 1 | 3 |  |  | 1 | 1 | TCT-motif | |
| 1 |  |  |  | 1 |  |  |  | I-box | |
|  | 2 |  | 1 |  |  |  |  | TCCC-motif | |
|  | 1 |  |  |  |  |  |  | ACE | |
|  | 3 | 1 |  | 1 |  |  | 1 | Box II | |
|  | 1 |  |  |  |  | 1 |  | 3-AF1 binding site | |
|  |  | 1 |  |  |  | 1 |  | ATCT-motif | |
|  |  | 1 |  |  |  |  | 1 | AE-box | |
|  |  | 3 |  |  |  |  |  | AT1-motif | |
|  |  | 1 |  |  |  |  |  | ATC-motif | |
|  |  | 1 |  |  |  |  | 1 | ACE | |
|  |  |  | 1 | 1 |  |  |  | GATA-motif | |
|  |  |  |  | 1 | 1 |  | 1 | LAMP-element | |
|  |  |  |  |  | 1 |  |  | GATT-motif | |
| 1 |  |  |  |  |  |  |  | TC-rich repeats | Abiotic stress |
|  |  | 1 |  |  |  |  |  | WUN-motif | |
|  | 2 |  |  |  |  |  | 1 | O2-site | |
| 1 | 5 | 3 | 1 | 2 |  |  | 2 | ARE | |
|  |  |  |  |  | 1 |  |  | GC-motif | |
| 1 |  | 1 |  |  | 1 |  |  | LTR | |
|  | 1 |  |  | 2 | 3 | 1 |  | MBS | |
| 2 |  |  | 2 |  | 1 |  |  | TCA-element | Phytohormone |
| 5 | 13 | 6 |  | 5 | 2 | 3 | 6 | ABRE | |
| 3 | 3 | 1 | 2 |  | 2 | 2 | 1 | CGTCA-motif | |
| 3 | 3 | 1 | 2 |  | 2 | 2 | 1 | TGACG-motif | |
| 1 |  |  |  |  |  |  |  | GARE-motif | |
|  |  |  | 1 | 2 |  |  |  | TATC-box | |
|  |  |  |  |  |  |  | 1 | P-box | |
| 1 |  | 2 |  | 1 |  | 1 |  | TGA-element | |
|  | 1 |  |  |  |  |  |  | AuxRR-core | |

**Figure 5** **The number of *cis*-acting elements in *CisTPS* genes.** The figure was plotted based on the presence of *cis*-acting element responsive to light, abiotic stresses and phytohormones.

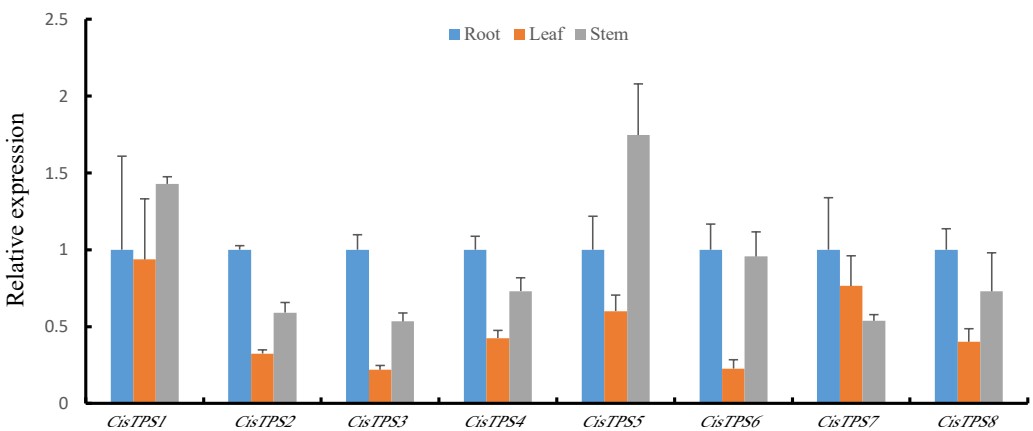

**Figure 6** Expression levels of *CisTPS* genes in root, leaf, and stem. Standard error bars represent standard error of the mean (SEM) of three replicates. Values are means ±SEM of three biological replicates.

## DISCUSSION

*TPS* genes play important roles in plant growth, development, and response to biotic and abiotic stresses (*Kosar et al., 2019*; *Paul, Watson & Griffiths, 2020*). Therefore, the *TPS* gene family has received more attention and has been identified in many plants. In this study, eight *TPS* genes were identified from the sweet orange genome. All the CisTPS proteins contained both a TPS domain at the N-terminus and a TPP domain at the C-terminus, indicating that the structures of the two domains might be formed before the differentiation of these members and is essential for *TPS* functions. Our results were consistent with research in *A. thaliana*, pepper, and apple (*Yang et al., 2012*; *Wei et al., 2016*; *Du et al., 2017*). While the TPP domain is missing in cotton *GrTPS6*, *GhTPS4*, and *GhTPS9* genes (*Mu et al., 2016*), perhaps due to evolution. The number of *TPS* genes varies greatly among species. For example, there are 53 *TPS* genes in cotton, 31 in *Brassica napus*, 15 in cabbage, 14 in Chinese cabbage, 12 in winter wheat, 11 in *A. thaliana*, and seven in grapevine (*Leyman, Dijck & Thevelein, 2001*; *Xie et al., 2015a*; *Mu et al., 2016*; *Morabito, Secchi & Schubert, 2021*; *Zhou et al., 2021*). These results indicated that the *TPS* gene family was not conserved in different species.

Eight *TPS* genes from sweet orange were divided into two subfamilies based on the amino acid sequences, as previously observed in cucumber, tree peony, *Populus*, *A. thaliana* and rice (*Dan et al., 2021*; *Leyman, Dijck & Thevelein, 2001*; *Sun, Chen & Tao, 2021*; *Yang et al., 2012*; *Zang et al., 2011*). *CisTPS1* and *CisTPS7* belonged to Class I, whereas the remaining six genes (*CisTPS2*, *CisTPS3*, *CisTPS4*, *CisTPS5*, *CisTPS6*, and *CisTPS8*) belonged to Class II. AtTPS proteins of Class I encoding catalytically active enzymes showed TPS activity (*Blázquez et al., 1998*; *Vandesteene et al., 2010*), indicating the TPS activity of CisTPS1 and CisTPS7. Furthermore, *CisTPS7* and *AtTPS1* clustered together, implying that *CisTPS7* may have functions similar to *AtTPS1* and may play a crucial role in sweet orange growth, development, and stress response. Although the *AtTPS* genes of Class II lacked TPS and TPP activities, they were preserved under evolutionary selection pressure and differed

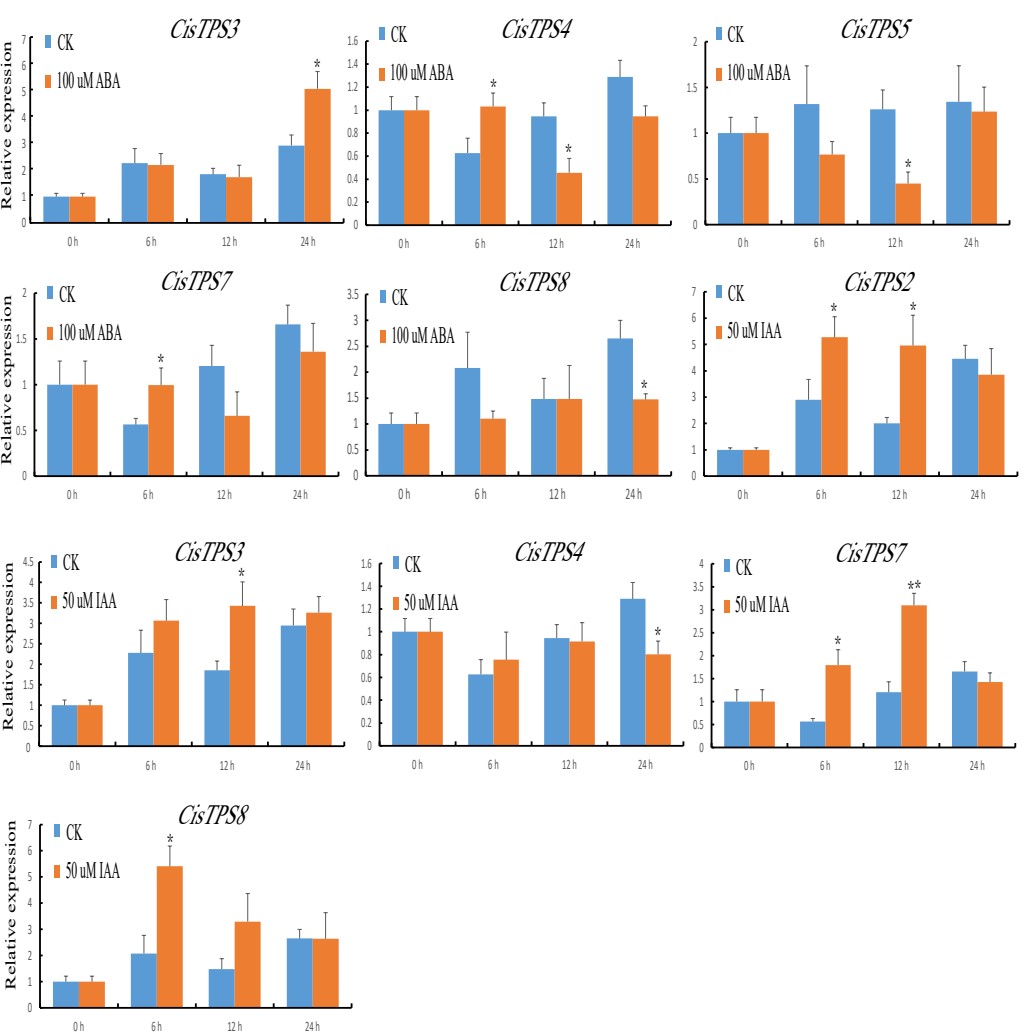

**Figure 7** **Expression levels of *CisTPS* genes induced by phytohormones.** Seedlings were treated with 100 μM ABA and 50 μM IAA. Standard error bars represent standard error of the mean (SEM) of three replicates. Values are means ±SEM of three biological replicates. Asterisks indicate statistical significance determined by Student's *t*-test (* $P < 0.05$, ** $P < 0.01$).

in tissue and expression rate, suggesting that they had particular functions (*Zang et al., 2011*). However, the activities and functions of most Class II members remain uncertain. Although the activities and functions of Class II *CisTPS* genes cannot be determined in this study, we could speculate that they may perform some specific functions.

Exon–intron diversification plays a major role in diverse gene family evolution (*Qi et al., 2020*). Previous studies have shown that the *TPS* genes in Class I mainly contained 16 introns and those in Class II mostly harbored two introns (*Yang et al., 2012*). Class I genes in sweet orange contained 16–17 introns, and Class II genes had two introns except for *CisTPS3* and *CisTPS6*. Based on the analysis of the *CisTPS* gene structure, the number of exons and introns of Class I genes was pronouncedly higher than that of Class II, and

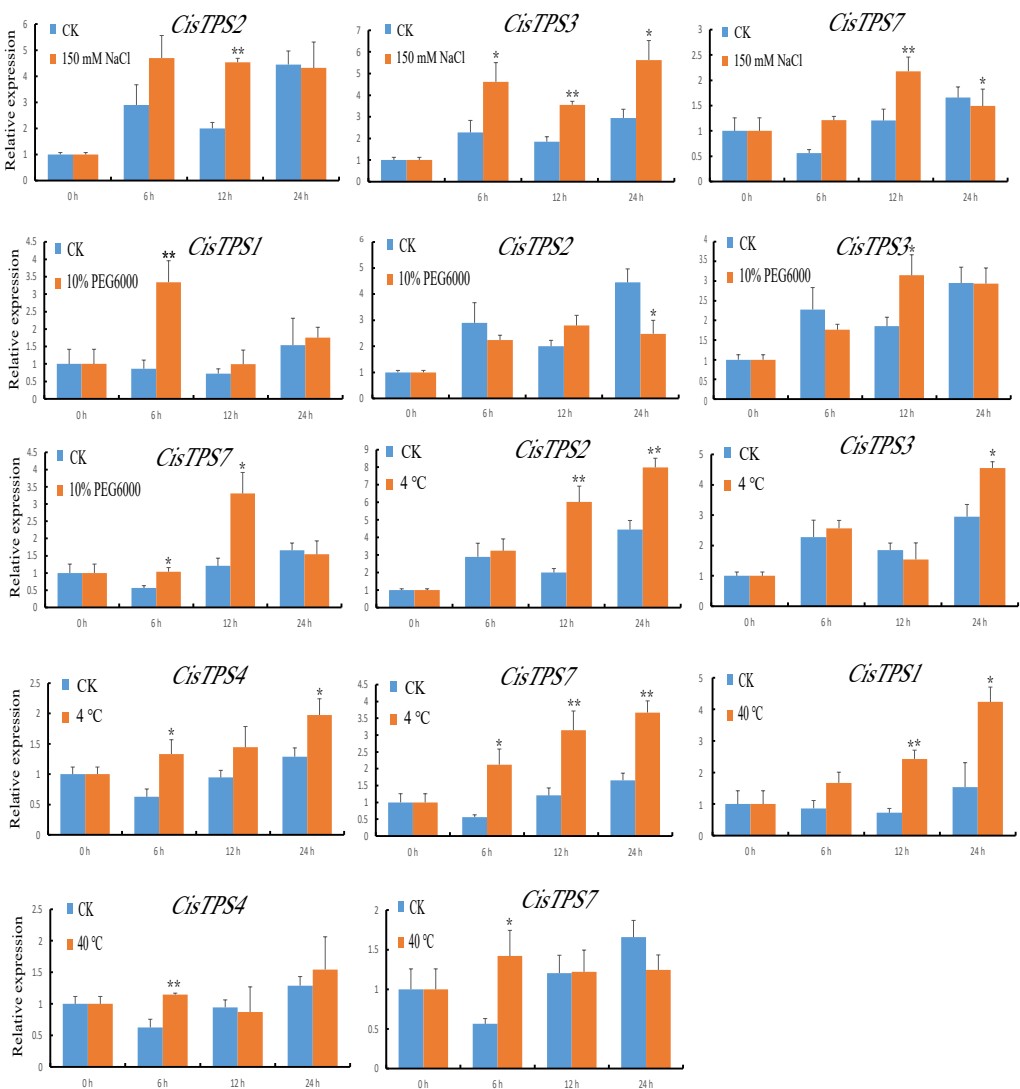

**Figure 8** **Expression levels of *CisTPS* genes induced by abiotic stresses.** Seedlings were treated with 150 mM NaCl, 10% PEG-6000, low temperature (4 ° C) and high temperature (40 ° C). Standard error bars represent standard error of the mean (SEM) of three replicates. Values are means ±SEM of three biological replicates. Asterisks indicate statistical significance determined by Student's *t*-test (* *P* < 0.05, ** *P* < 0.01).

almost the same number of exons and introns was present in the same class. Class I and II genes experienced distinct selection pressures and evolutionary processes, and Class II genes lost some introns because of strong selective pressure during evolution (*Zhaxybayeva & Gogarten, 2003*). Moreover, the results showed that gene evolution was consistent with conservation in gene family structure, as it did for *A. thaliana*, rice, and winter wheat (*Lunn, 2007*; *Zang et al., 2011*; *Xie et al., 2015a*).

Motif analysis showed that 20 dissimilar conserved motifs were obtained in the *CisTPS* gene family. In Class I, both *CisTPS1* and *CisTPS7* had 15 motifs and lacked motif 8, 10,

15, 18, 19, and 20. In Class II, four *CisTPS* genes contained all 20 motifs, but *CisTPS5* was deficient in motif 18 and 20, and *CisTPS6* lacked motif 18. Thus, 15 motifs were observed in all *CisTPS* genes. Therefore, the 15 motifs may be crucial for *CisTPS* genes to maintain their structure and function. In addition, gene sequences closer in the phylogenetic tree showed highly similar motifs.

*Cis*-acting elements control gene expression by combining with activated transcription factors when plants were under stress (*Hadiarto & Tran, 2011*). In the promoter regions of *CisTPS* genes, the *cis*-acting elements were related to environmental stress (light, oxygen concentration, temperature, drought, and wounding) and exogenous phytohormones (salicylic acid, ABA, MeJA, auxin, and gibberellin). These elements were found in *Populus* and cucumber (*Gao et al., 2021*; *Dan et al., 2021*), suggesting that *TPS* genes regulate stress, phytohormone, and light responses.

The gene expression difference in different tissues may explain their vital roles in specific tissues. *CisTPS* genes were detected in roots, leaves, and stems, and these results were in agreement with the results of cucumber and *Medicago truncatula* (*Dan et al., 2021*; *Song et al., 2021*). The expression of *CisTPS2*, *CisTPS3*, *CisTPS4*, *CisTPS6*, and *CisTPS7* was the highest in roots and that of *CisTPS1* and *CisTPS5* was the highest in stems. Based on the results, *CisTPS2*, *CisTPS3*, *CisTPS4*, *CisTPS6*, and *CisTPS7* may be involved in root development, whereas *CisTPS1* and *CisTPS5* may mediate stem development.

Trehalose is important for higher plants to preserve bioactive substances and cell structures when faced with damaging environmental stresses (*Garg et al., 2002*; *Jang et al., 2003*). Consequently, the expression level of *TPS* genes from some plants has been tested under different stress conditions (*Iordachescu & Imai, 2008*; *Xie et al., 2015a*; *Mu et al., 2016*; *Morabito, Secchi & Schubert, 2021*; *Song et al., 2021*). In this research, *CisTPS7* responded to every treatment, especially to IAA, salt, and low temperature treatment. Based on the phylogenetic analysis, we have found that the *CisTPS7* gene corresponds to the *AtTPS1* gene. *A. thaliana* seedlings overexpressing *AtTPS1* displayed dehydration tolerance and ABA-insensitive phenotypes (*Avonce et al., 2004*). *OsTPS1* overexpression in rice conferred seedling tolerance to cold, salt, and drought stresses (*Li et al., 2011*). Transgenic potato plants of the *TPS1* gene from *Saccharomyces cerevisiae* clearly increased drought resistance (*Yeo et al., 2000*). For that reason, we assert that *CisTPS7* may play a valuable role in sweet orange stress resistance. In addition, the Class I members in red algae were significantly upregulated under high temperature and desiccation (*Sun et al., 2019*). In agreement with the results, the expression of Class I genes (*CisTPS1* and *CisTPS7*) increased in response to drought and high temperature stresses.

The Class II members reveal different expression patterns under various stresses. Overexpression of Class II *OsTPS* genes enhanced rice tolerance to abiotic stress (*Li et al., 2011*). The transcript level of *AtTPS5*, a negative regulator in ABA signal transduction, was elevated during heat stress (*Suzuki et al., 2008*; *Tian et al., 2019*). *CisTPS4* corresponding to *AtTPS5* responded to ABA, IAA, and cold and heat stresses. *AtTPS7* expression increased under salt stress (*Renault et al., 2013*). *CisTPS2*, homologous with *AtTPS7*, was strongly induced by salt stress and low temperature. *AtTPS9* was significantly upregulated by treatment conditions such as ABA, salt, drought, and high temperature (*Suzuki et al., 2008*).

However, *CisTPS8* in sweet orange was the most homologous to *AtTPS9*, and it was only slightly induced by phytohormone (ABA and IAA) stresses. Furthermore, *CisTPS3* was observed to be induced by multiple stresses, such as ABA, IAA, salt, drought and cold, whereas *CisTPS5* was only repressed by ABA. In sweet orange, *CisTPS6* showed no response to various treatment conditions, which also existed in the soybean *TPS* gene family (*Xie, Wang & Huang, 2014*).

## CONCLUSIONS

To understand more about the *TPS* gene family in *C. sinensis*, eight *CisTPS* genes were identified from the sweet orange genome in this study. The *CisTPS* genes were located on five chromosomes and were divided into two subfamilies-Class I and II. *CisTPS* genes were similar to *AtTPS* genes in their conserved domain and gene structure. In addition, most *CisTPS* genes responded to phytohormones and abiotic stresses, and six *CisTPS* genes were even controlled by multiple stresses. The results indicated that *CisTPS* genes were required for the response to phytohormones and abiotic stresses in sweet orange. Our findings provide basic resources for further studies of the functions of the *TPS* gene family on stress-resistance, growth, and development in sweet orange.

## ACKNOWLEDGEMENTS

We would like to thank TopEdit for the linguistic assistance during the preparation of this manuscript.

### Funding

This work was supported by the National Key R & D Program of China (2021YFD1400800). The funders had no role in study design, data collection and analysis, decision to publish, or preparation of the manuscript.

### Grant Disclosures

The following grant information was disclosed by the authors:
The National Key R & D Program of China: 2021YFD1400800.

### Competing Interests

The authors declare there are no competing interests.

### Author Contributions

- Kehong Liu conceived and designed the experiments, performed the experiments, analyzed the data, prepared figures and/or tables, authored or reviewed drafts of the article, and approved the final draft.
- Yan Zhou conceived and designed the experiments, analyzed the data, authored or reviewed drafts of the article, and approved the final draft.

## Data Availability

The raw data are available in the Supplemental Files.

## Supplemental Information

Supplemental information for this article can be found online at http://dx.doi.org/10.7717/peerj.13934#supplemental-information.

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
