# Peer review of "Genome-wide identification of the trehalose-6-phosphate synthase gene family in sweet orange (Citrus sinensis) and expression analysis in response to phytohormones and abiotic stresses"

_PeerJ, doi:10.7717/peerj.13934_

## Round 0.1 · original submission · Minor Revisions

Authors have shown a well-structured manuscript.

Reviewer 1 ·

Basic reporting

Language and format needs to be perfected.

Experimental design

1 Figure 3 need to be perfecte, for example Cis TPS need to highligh, etc.. There are few species selected, please increase.
2 In addition to the prediction of cis-acting elements, the author should clone promoter of CisTPS and further analyze them to get more accurate results.

Validity of the findings

The significance of the study is not sufficient, please add or modify.

Additional comments

In the stress treatment, what is the basis for the choice of stress conditions, time, phytohormone, etc.?  Please add it.

Reviewer 2 ·

Basic reporting

The manuscript entitled “Genome-wide identification of the trehalose-6-phosphate synthase gene family in sweet orange (Citrus sinensis) and expression analysis in response to phytohormones and abiotic stress” describes the in silico identification of 8 TPS genes within C. sinensis genome. The article is complemented with an experiment for testing the hypothesis of homology function of these genes by checking their expressions under biotic and abiotic stresses. The language is clear, and the authors provided a sufficient literature background. The article is well-structured, consistent results were shown, and the raw data is provided. The figures are relevant for the content of the article, however most of them lack of information, resolution and any figure legend were presented.
Here are my suggestions/questions for further improvement of the manuscript:
Line 30/93: What does the expression “common transgenic material” means? Please, rephrase it.
Line 106-107: How was the criteria for the selection of “candidate TPS protein sequences” ? Please, explicit the method used for retrieving the protein sequences in the genome.
Line 146: Please, specify which variety/accession was used in the experiments.
Line 183: Please, remove “i.e.”
Line 200: In Figure 2, only 5 are shown.
Line 204: watermelon??
Line 244-245: It is too strong to affirm that these play an essential role in stress response. No molecular confirmation is showed in the manuscript.
Line 293-294: How is TPS subfamilies are divided in the other cited species? There are only two subfamilies as in Arabidopsis and rice?
Line 298: This is indicating a probable TPS activity.
Line 300: … and may play a crucial role.
Line 301-304: Please relate this information provided for Arabidopsis to your found results for citrus.
Line 348: CisTPS may play a valuable role.

Experimental design

The experimental design is compatible with the presented results.
The Material & Method would be improved if the “Plant materials and treatment conditions” section were divided by applied stress/experiment, making explicit each control. This section is a little confusing.
The figures should be further improved. Any legend was presented. The resolution is of poor quality in general, and some words and numbers are presented in really small size. Some basic information is lacking in some figures (i.e., axis titles in Fig. 7).
Figure 2: Please indicate the name of the CisTPS genes in the figure.
Figure 4: It would be interesting to indicating the subdivision of Class I et Class II TPS proteins A.
Figure 6: Data points should be shown in conjunction with barplots. (Barplots can be misleading).
Figure 7: Font size is too small, and axis title is missing. Data visualization would be improved if data is presented as time-serie plot instead of barplots.
Figure 8: Idem of Fig. 7. Why the data for all genes were not shown? The figure would be improved if a clear separation for stresses is presented.

Validity of the findings

The findings are interesting. The validation experiments make the results robust and consistent, bringing insights of the role of these genes beyond the in silico approach. The authors should be more careful, though, when presenting the results because the experiments shown are not functional validation.

·

Basic reporting

I have reviewed the manuscript: "Genome-wide identification of the trehalose-6-phosphate synthase gene family in sweet orange (Citrus sinensis) and expression analysis in response to phytohormones and abiotic stresses" , and I think in general that the authors have done a good job.

The study presents the results of primary scientific research, and are not reported elsewhere.

Without doubt, the purpose of this manuscript has a very interesting.

Experimental design

No comment.

Validity of the findings

The present study brings new information about the role played by the trehalose-6-phosphate synthase gene family in a species of agronomic importance.

Additional comments

Dear authors,

I congratulate everyone for the description of this study. However, I suggest taking care when making certain citations. Try to use current and relevant references for the proposed subject.

---

## Round 0.2 · accepted · Accept

The authors addressed most of the referees' issues.

Reviewer 1 ·

Basic reporting

No comment.

Experimental design

The authors reponsed to most of my comment. there a little regretful that the author did not supplement the relevant experimental results of promoters and no explanation.

Validity of the findings

No comment.

Additional comments

No comment.

Reviewer 2 ·

Basic reporting

I reviewed the modifications for the manuscript entitled “Genome-wide identification of the trehalose-6-phosphate synthase gene family in sweet orange (Citrus sinensis) and expression analysis in response to phytohormones and abiotic stress”. After a first round of revisions, the authors provided a clarification and modifications for most of my questions raised after my first reading.

Experimental design

No comments

Validity of the findings

No comments